



# Data for Modern Soil Chronometry using Fallout Radionuclides

Joshua D. Landis[1]

[1]Dept. Earth Sciences, Dartmouth College, 19 Fayerweather Hill Road, Hanover, NH USA 03755

*Correspondence to*: joshua.d.landis@dartmouth.edu

**Abstract**

We describe environmental gamma spectrometry data for >700 soil samples collected from >35 high-resolution quantitative soil profiles spanning global sites. The data are collected for the purpose of modern soil chronometry based on fallout radionuclides (FRNs) [7]Be and [210]Pb, using the Linked Radionuclide Accumulation model (LRC). Cumulative gamma counting time for samples in the database exceeds 6.5 years. This is a living database to be augmented as data become available and corrected with
improvements in data reduction or identification of errors. Versions and changes will be indexed. Special attention is paid to measurement uncertainties in the dataset, as well as how atmospheric or excess [210]Pb is defined in both geochemical and mathematical terms for use in the LRC model. Basic familiarity with gamma spectrometry and radionuclide decay chains is assumed. The data set can be accessed at https://doi.org.10.17632/cfxkpn6hj9.1 (Landis, 2025).


**Short Summary**

Understanding rates of environmental change is critical to human and ecological health but is difficult when the processes are too slow or too small to observe directly. To overcome this limitation, we can use
natural radioactive elements as virtual 'clocks' to measure change. Here we describe a large number of measurements that have been used to develop soils as clocks or chronometers of change to atmospheric carbon and mercury (Hg) cycles.

**1. Introduction**

We describe high-precision measurements of environmental radionuclides collected for the purpose of developing modern soil chronometry. Numerous problems such as atmospheric pollutant deposition and carbon sequestration require knowledge of the rates of soil processes that influence metals and carbon through time, but an appropriate chronometry system spanning years to centuries of environmental
change has not been available until recent efforts to validate fallout radionuclide (FRN) chronometry (Landis, 2023; Landis et al., 2016, 2025; Landis, Obrist, et al., 2024; Landis, Taylor, et al., 2024). Historically, the impediment to adoption of FRNs in soil systems has been the availability of sufficient instrumental resources to provide long counting times that are necessary for measurement of short-lived [7]Be, in datasets large enough to develop appropriate age models (Kaste et al., 2011). Here, we report the
fallout radionuclides [7]Be and [210]Pb, together with bomb pulse [241]Am and [137]Cs, and ancillary radiogenic isotopes including [226]Ra, [228]Th, [228]Ra, and others, that are used to construct and verify the Linked Radionuclide Accumulation model (LRC). These radionuclides are all measured concurrently by gamma spectrometry in the Fallout Radionuclide Analytics facility (FRNA) at Dartmouth College. The data are accessible at https://doi.org.10.17632/cfxkpn6hj9.1; Landis, 2025).


LRC is an empirical age model that dates soil organic and mineral matter based on a mass balance comparison between the soil accumulative [7]Be:[210]Pb ratio ($R_{acc}$) and the [7]Be:[210]Pb atmospheric flux ratio





$(R_D)$(Landis et al., 2016). The data presented currently include more than 700 samples from over 35 high-resolution quantitative soil pits collected using field methods adapted for this purpose, and more are being added each month (Fig. 1). The method is confirmed by accurately dating the bomb-pulse peak in $^{241}$Am (or $^{137}$Cs where appropriate) which is known to have occurred in 1963-64 (Health and Safety Laboratory, 1977). The method is further corroborated by both novel $^{228}$Th:$^{228}$Ra chronometry and $^{14}$C approaches and yields uncertainties on the order of 10% up to 100 years and 20% up to 200 years (Landis, 2023; Landis et al., 2025).

The majority of sampled soils are what we refer to as 'reference', selected to minimize the potential for historical disturbance through development, land use, natural erosion or tree throw over the past minimum of 150 years. Additional sites may include experimental designs examining hillslope and alluvial or fluvial processes and should not be taken as representative of reference conditions, i.e. with an expectation that the nuclear bomb-pulse can be accurately dated. Implementation of the LRC model is described elsewhere (Landis et al., 2016). Additional details and challenges of environmental gamma spectrometry (Cutshall et al., 1983; Landis et al., 2012; Murray et al., 1987, 2018) or theories of related FRN age models can also be found elsewhere (Arias-Ortiz et al., 2018; Barsanti et al., 2020; Sanchez-Cabeza & Ruiz-Fernández, 2012). Any brief discussion of these topics here is intended only to contextualize the data and thereby provide an entry point for non-specialist readers. A basic understanding of gamma spectrometry and radionuclide decay chains is assumed.

Finally, data validation is discussed with respect to both analytical problems such as instrument calibration, as well as geochemical problems, namely the susceptibility of Rn to diffuse out of samples in the laboratory. A priority of this manuscript is to demonstrate that routine environmental gamma spectrometry is capable percent-level precision that is expected from other capital analytical platforms such as inductively-coupled plasma optical or mass spectrometry used for elemental and isotopic analyses (ICPOES, ICPMS).

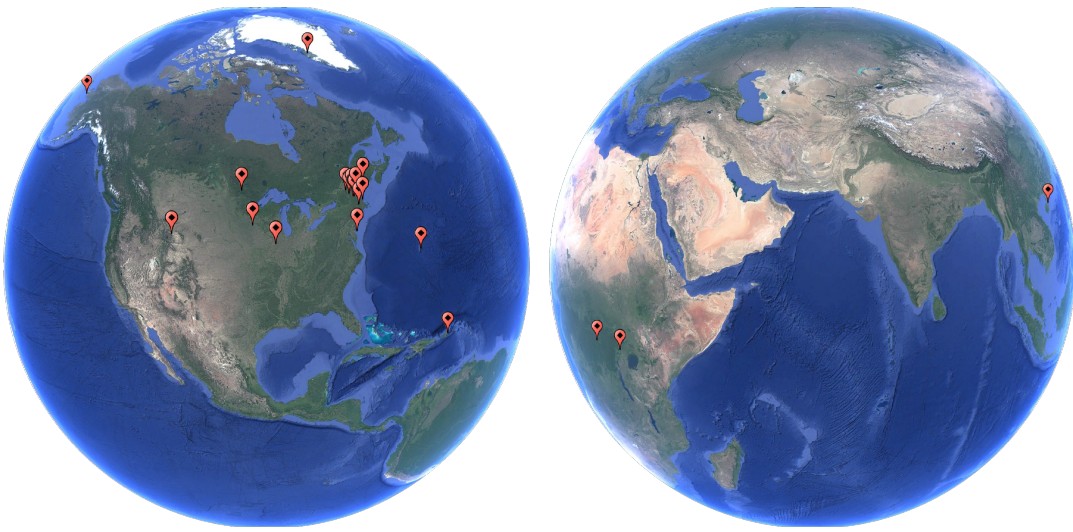

**Figure 1. Site map for soils collected for modern soil chronometry (© Google Earth Pro, data SI, NOAA, U.S. Navy, NGA, GEBCO. Image IBCAO, Landsat, Copernicus. Altitude 5181 miles).**



## 2. Methods

### 2.1. Instrumentation

Data are collected with six Mirion (formerly Canberra) Broad Energy intrinsic germanium detectors (BEGe) with carbon fiber endcaps and ultra-low background cryostats which are cooled with liquid nitrogen. The detectors are shielded with 4 inches of ultra-low background lead. Data acquisition is performed with Mirion DSA-1000 or Lynx digital signal processors running with Genie 2K software v.3. The specific BEGe detector design is important because it provides excellent efficiency at low gamma energies for $^{210}$Pb acquisition without sacrificing efficiencies at high energies. Carbon fiber provides good transmission of the low energy $^{210}$Pb emission which is otherwise attenuated by an aluminum endcap. Samples are typically run using four-day count times. Of two detector models, BEGe 3830 (38 cm$^2$ detector active area, 30 mm thickness) and larger BEGe 5030 (50 cm$^2$), the latter saves one day of counting and accomplishes in three days what the 3830 accomplish in four.

### 2.2. Sample preparation and geometry

Foliage, litter, and soil samples are oven dried at 60 °C. Bulk soils are sieved or hand-picked to remove clasts and debris >2mm in diameter. Fine roots >1 mm diameter or >2 cm in length are removed by hand. Foliage and leaf litter (Oi and Oe horizon) are typically milled to <1 mm using a Wiley mill with stainless steel blades. Soil samples are loosely disaggregated as necessary by agate mortar and pestel and packed into 110 cm$^3$ polyethylene petri dishes with snap closures. The petris are sealed with common electrical tape and rested for >2 weeks to allow for $^{222}$Rn ingrowth, although we will show that this approach does not quantitatively retain Rn (see also (Murray et al., 1987, 2018)). Our handling of the problem of Rn emanation from both laboratory samples and soils *in situ* is discussed below.

### 2.3. Calibration and Sample Self-Absorption

Gamma detector calibration is based on geometric efficiency, which is the proportion of photons emitted by the sample that are recorded by the detector, absent sample self-absorption of photons which is dependent on its specific elemental composition. Net efficiency includes self-absorption of photons by the sample. We measure net efficiency directly by using a series of soils/sediment doped 1% by weight with either uranium ore BL5 or thorium ore OKA2 which are available from Natural Resources Canada Certified Reference Material Project. Geometric efficiency for the standards is then calculated as the quotient of net efficiency and the standard self-absorption coefficient at each photopeak energy in the (described below). The efficiencies are representative of the detector-sample geometric relationship and applicable to all sample unknown measurements. Efficiencies for photopeaks not represented in the standareds are estimated by regression from adjacent photopeaks in the U or Th standards (e.g., $^7$Be, $^{241}$Am, or $^{137}$Cs; these are not known to have coincidence summing effects). Reported uncertainties on the standard activities is 0.2% for U and 1.0% for Th, and secular equilibrium is assumed for both.

Net efficiency for individual samples is calculated as the product of geometric efficiency determined from standards and the sample self-absorption at each energy of interest (Landis et al., 2012). To do this, self-absorption for each sample is measured using a planar uranium ore source (same diameter as sample) placed directly on top of the sample. The is performed as a second data acquisition following the sample 'long count' which does not use point source. Self-absorption at all energies is calculated using the method of (Cutshall et al., 1983), or in the case of non-uranium series radionuclides, is estimated by linear regression from the U-series photopeaks.

Calibration uncertainty estimated from geometric efficiencies of 6 standards of bulk density ranging from 0.5 g cm$^{-3}$ to 200 g cm$^{-3}$ is less than 2% relative standard deviation for each of our detectors. Uncertainty



of sample self-absorption derived from peak integration (below) contributes <1%. Thus, our net efficiency calibration uncertainties for most radionuclides are <2% at the one-sigma level.


### 2.4. Peak integration

A semi-automated peak integration routine is implemented in Excel as described by (Landis et al., 2012). For each detector, peak shape parameters are measured across energies either within the collected sample spectrum using peaks >10,000 counts or using prior collection of a high-activity point source

spectrum. A peak shape model is then used to estimate the partial area of sample peaks that are integrated digitally by summation. Partial peak integration reduces measurement uncertainty by avoiding peak tails which contribute disproportionately to integration error.

### 2.5. Spectral Background Correction

All radionuclide peaks are corrected for background contributions which are measured with composite background spectra totaling hundreds of days of count time. Background for each peak may variously include ambient Rn in the laboratory, artifacts of the gamma spectrum, impurities in the detector housing and shielding, and ambient construction materials in the laboratory facility.

### 2.6. Data Quality and Verification

Environmental gamma spectrometry is capable of producing percent-level precision on par with other capital analytical platforms such as ICP-MS. Precision in gamma spectrometry estimates of radionuclide activity is easily documented because many radionuclides emit photons at multiple energies. These typically have variable coincidence summing effects that require correction at each energy, as well as

sample self-absorption factors that also require correction at each energy. Concordance among estimates from multiple photopeaks spanning the energy spectrum thus provides a robust estimate of precision in the gamma measurements.

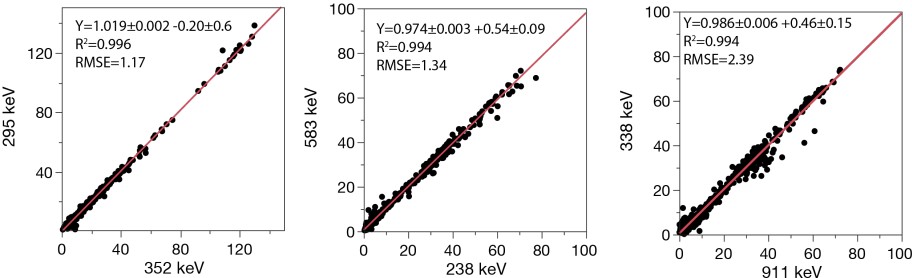

**Figure 2. Correlations of multiple photopeaks for (a) $^{222}$Rn, (b) $^{228}$Th, (c) $^{228}$Ra. Lines show linear best fits.**

For quality control measures we recommend reporting multi-line means and uncertainties for three radionuclides: $^{222}$Rn measured at $^{214}$Pb (295, 352 keV) and $^{214}$Bi (609 keV); $^{228}$Ra measured with up to four emission lines from $^{228}$Ac (209 keV, 338 keV, 463 keV, 911 keV); and $^{228}$Th measured by $^{212}$Pb (238 keV),

$^{208}$Tl (538 keV), and possibly $^{224}$Ra (241 keV) (Fig. 2).

The coefficient of variance (CV) or relative standard deviation (RSD) for multiple emission peaks for each of $^{222}$Rn, $^{228}$Th, and $^{228}$Ra provides an assessment of measurement accuracy and precision in the gamma spectrum (Fig. 3). These are typically in the 1-5% range for typical crustal activity concentrations of 20 Bq

kg$^{-1}$. $^{222}$Rn is importance since it is often used as an indirect measure of $^{226}$Ra with integration uncertainties 5-10 times lower (Fig. 3a), but with the important caveat that the actual values of the two measures differ substantially due to geochemical disequilibrium caused by Rn leakage (*Sect. 3.5*).

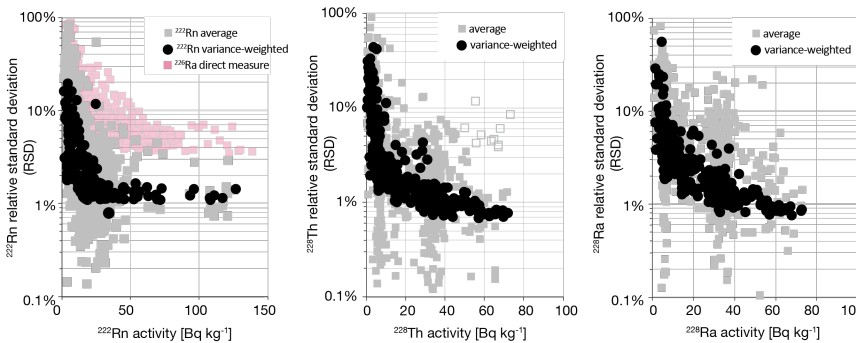

**Figure 3. Average and variance-weighted average activities for radionuclides measured at multiple energies in the gamma spectrum, including (a) $^{222}$Rn, with comparison to direct measure of $^{226}$Ra; (b) $^{228}$Th; (c) $^{228}$Ra. $^{226}$Ra measurement is possible at a single emission line which is compromised by low photon yield and photopeak convolution with $^{235}$U, resulting in uncertainties 5-10 times higher than possible with e.g. $^{222}$Rn daughter radionuclides.**

### 2.6. Peak interference deconvolution

Multiple important radionuclides in the gamma spectrum suffer significant peak-peak overlap that requires deconvolution. Most notably are $^{7}$Be which has interference from both $^{228}$Ac (478 keV) and $^{214}$Pb (280 keV) (Landis et al., 2012), and $^{226}$Ra which has interference from $^{235}$U at 185 keV and which approximately doubles its analytical uncertainty (Zhang et al., 2009). Our $^{7}$Be deconvolution regime is verified elsewhere using repeat measures of a single natural sample since the known half-life of $^{7}$Be can be exploited to predict its change in concentration over monthly timescales (Landis et al., 2012). The $^{226}$Ra deconvolution regime can be tested against certified reference materials; however, the reported uncertainties for $^{226}$Ra in available standards typically exceeds 10% and are therefore of limited use in verifying robust measure of this radionuclide (more below in *Sect. 8*).

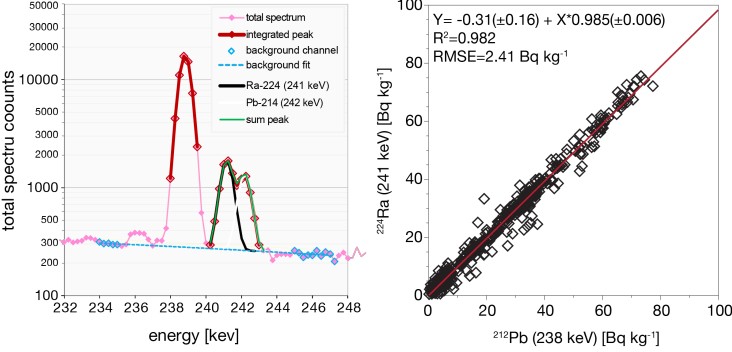

**Figure 4. (a) Measurement of $^{224}$Ra is possible in the gamma spectrum but requires deconvolution of its 241 keV emission from $^{214}$Pb at 242 keV. (b) Accuracy of the deconvolution procedure can be tested against $^{212}$Pb (via $^{220}$Rn) measured at 238 keV.**

Deconvolution can also be tested using the measure of $^{224}$Ra (half-life 3.6 days, 241 keV) as a proxy for its parent radionuclide $^{228}$Th because of its photopeak overlap with $^{214}$Pb at 242 keV. The peak deconvolution procedure is evaluated with the expectation that radioactive equilibrium is maintained, i.e., with a slope of 1 in correlation between the parent and daughter (Fig. 4). Here the slope =0.985 ±0.006 which shows a systematic error of just 1.5 ±0.6%.

### 3. Radionuclide measurements

*3.1. $^7$Be and problem of detection.* The defining limitation in LRC development is the problem of $^7$Be
detection. The combination of low natural abundance, low photon yield (10.5%), and short half-life (54
days) requires extraordinary efforts in both the logistics of sample collection and availability of
instrument time to make these measurements possible. Samples should be measured within 1 month of
collection and must be counted for 4 days or more per sample to define the $^7$Be soil depth profile (Fig. 5).
Following these protocols $^7$Be may be detected to depths of 10 or more centimeters in humid or moist
climates, which has important implications both for how soils are perceived to function and how age
models are applied to soils (Landis et al., 2016).

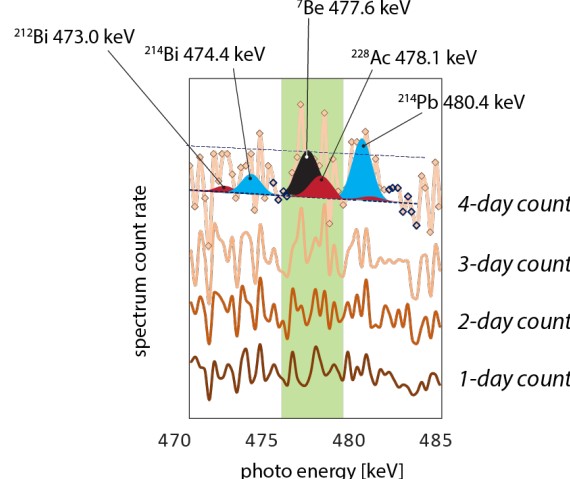

**Figure 5. Elapsed measurement of  the $^7$Be gamma photoregion over 1, 2, 3, and 4-day count times. The
measured sample is soil from 5 cm depth, and $^7$Be activity is quantified as 1.8 ±1.0 Bq kg$^{-1}$ after 4 days. The**
**modeled $^7$Be peak is shown in black, U-series peaks in blue, and Th-series peaks in red. MDL is shown with
dashed line.**

*3.2. $^{241}$Am and problem of detection.* Demonstrating accuracy of LRC requires concordance with
independent chronometers, and bomb-pulse $^{241}$Am is the prime candidate due to its long half-life,
limited solubility or bioavailability (in contrast to $^{137}$Cs), and potential for its concurrent measurement in
the gamma spectrum. The bomb-pulse has a well-known history of activity in the atmosphere (Appleby
et al., 1991; Health and Safety Laboratory, 1977). Abundance of $^{241}$Am is low, however, rarely exceeding 1
Bq kg$^{-1}$ in temperate soils, and is altogether undetectable in equatorial soils where fallout deposition was
10-times lower than mid-latitudes (Chaboche et al., 2021). Alternate measurement of Pu isotopes by
ICPMS or AMS is becoming routine and may offer an alternative where $^{241}$Am cannot be measured (e.g.
(Alewell et al., 2017)).

*3.3. $^{137}$Cs and problem of geochemical mobility.* In contrast to $^{241}$Am, $^{137}$Cs is easily measured in the
gamma spectrum. However, it can be subject to biological cycling as a K analog (Kaste et al., 2021). As a
result, the shape of Am and Cs profiles can diverge remarkably (Fig. 6). $^{241}$Am is always the preferred
measure of bomb pulse. Still, where $^{241}$Am and $^{137}$Cs profiles appear consistent, $^{137}$Cs may provide insight
into shape of the $^{241}$Am profile when latter is detection limited.

*3.4. Total ²¹⁰Pb.* Total $^{210}$Pb is the sum of both atmospheric (excess) and geogenic (supported) sources
and is easily measured in gamma spectrum with BEGe detectors. The strongest source of error is the self-
absorption correction since this can vary by a factor of 2-3 through the soil profile. However, the low
photon yield (4%) can make measurement of supported $^{210}$Pb challenging in soils with typical uranium
abundance. The four-day count time is thus important for providing adequate precision for defining
$^{210}$Pb$_{ex}$ at the tail of its depth distribution (where $^{210}$Pb$_{ex}$ is calculated as $^{210}$Pb$_{total}$ minus $^{222}$Rn; see below).


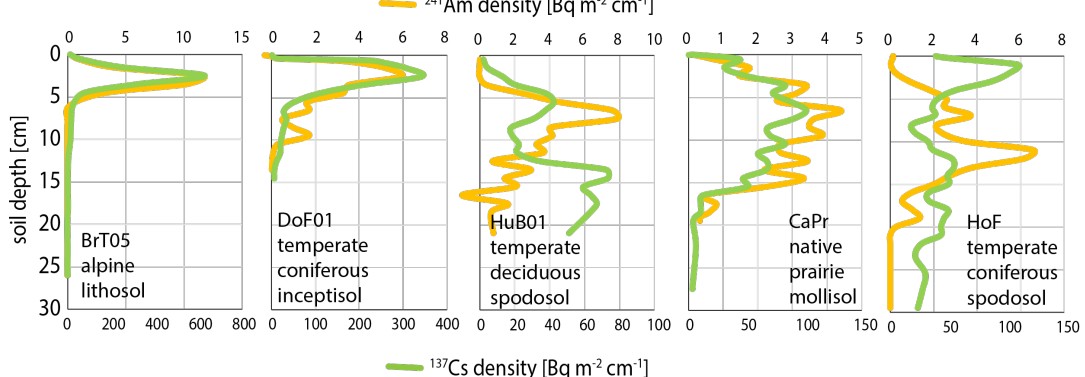

**Figure 6. Depth profiles of $^{241}$Am and $^{137}$Cs in diverse soils. In all cases the cumulative $^{241}$Am date is consistent
with  bomb-pulse peak of 1963-64. Assigned calendar years are as follows: BrT05 =1970.1, DoF01 =1965.3,
HuB01 =1964.4, CaPr01 =1964.2, HoF01 =1958.1. In some cases, $^{137}$Cs follows $^{241}$Am closely, in others (especially**
**spodosol) their divergence is pronounced and $^{137}$Cs follows a "leaky cycle" whereby some portion may be
strongly retained near the soil surface while any escaping this surface peak is rapidly flushed through the soil
profile below.**

*3.5. $^{226}$Ra and problem of radon emanation.* The measure of $^{226}$Ra is important, in principle, for defining
the activity of $^{210}$Pb that is supported in a soil by secular equilibrium with U-bearing mineral content
(discussed in detail in *Sect. 4*). Secular equilibrium requires that radionuclides in a decay chain have
equal activities, and we should observe this in the chain $^{226}$Ra > $^{222}$Rn >> $^{214}$Bi >$^{214}$Pb >> $^{210}$Pb provided
that gaseous Rn can be retained within the sample. Instead, however, we observe that $^{214}$Pb and $^{214}$Bi
underestimate $^{226}$Ra even in samples that are sealed and rested for the required ingrowth period for Rn
to achieve equilibrium (5 half-lives or about 20 days) (Fig. 7). Disequilibrium between $^{222}$Rn and $^{226}$Ra is
difficult to assess in a single sample, however, because of high uncertainties on the $^{226}$Ra measure due in
part to significant overlap from the primary $^{235}$U emission at 185.7 keV (Zhang et al., 2009).

In this dataset we quantify disequilibrium between $^{222}$Rn and $^{226}$Ra as *f*Rn, or fraction Rn, which is simply
the ratio of the two measures. (We acknowledge that the problem is somewhat more complicated
because the effective volume of Rn in the detector shield will be greater than volume of the sample itself
due to its diffusion from the sample container into surrounding air). Among all soil pits the mean
measured *f*Rn =79 ±1% (±RSE, *n*=30) (Fig. 7). Importantly, the variance of *f*Rn within a single pit is much
smaller than among pits [p<0.0001], which means that *f*Rn is a property of the geochemistry of
individual soils (with respect to hosting of Ra in primary vs. secondary minerals, grain size, susceptibility
to weathering, geological age, etc.).

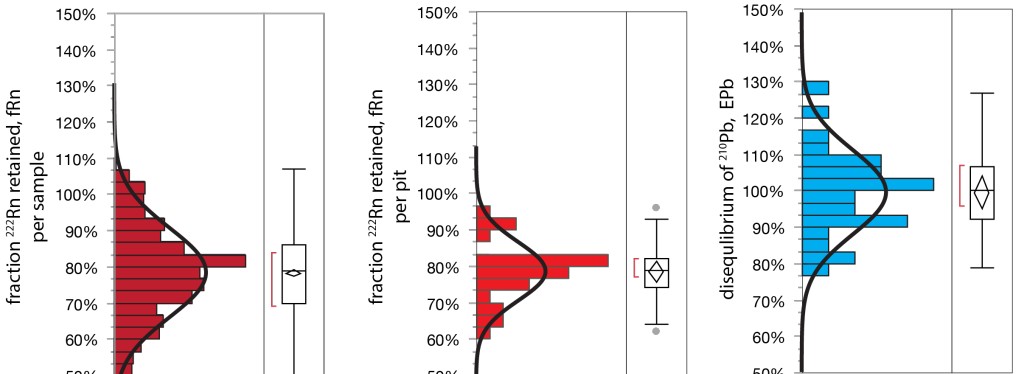

**Figure 7. Rn retention by soils, (a) fraction $^{222}$Rn retained per sample in laboratory, calculated as $^{222}$Rn/$^{226}$Ra, The mean rate of $^{222}$Rn disequilibrium in this dataset on a per-sample basis is 77 ±15% (SD, *n*=472, omitting samples with $^{226}$Ra <20 Bq kg$^{-1}$). (b) fraction $^{222}$Rn retained per soil pit, (c) laboratory disequilibrium between $^{210}$Pb and $^{222}$Rn in deep soils, per soil pit, calculated as εPb =$^{210}$Pb/$^{222}$Rn.**

### *3.6. Experimental confirmation of $^{226}$Ra accuracy and $^{222}$Rn losses.*

Affirming the accuracy of $^{226}$Ra measurements is foundational to interpreting $^{210}$Pb in the soil profile. Here we confirm the accuracy of direct $^{226}$Ra measurements (186 keV) against certified reference materials (CRMs) (Fig. 8), where performance is excellent with standard 4-day count times (Fig. 8a). We note that analytical uncertainties thus achieved are typically far better than those of the certification process itself. This reflects the difficulties in direct measure of $^{226}$Ra requiring corrections for self-absorption and deconvolution (Tasker et al., 2019).

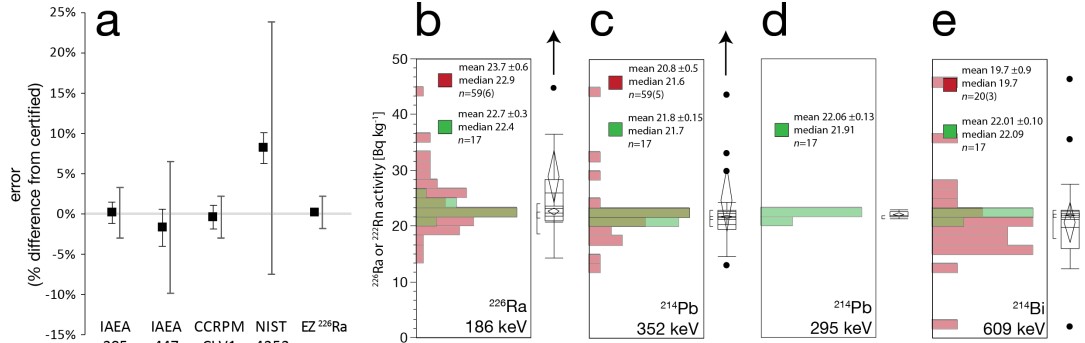

**Figure 8. (a) Certified reference materials for $^{226}$Ra determination, IAEA-385 (Irish sea sediment), IAEA-447 (moss-soil), CCRPM CLV-1 (spruce needle), NIST-4353a (Rocky Flats soil). Eckert and Ziegler single isotope solution (EZ $^{226}$Ra) is used as an in-house calibration material. Error bars are 1σ standard deviations for analysis at FRNA, larger error bars show certification uncertainties. (b-e) Distributions of results for IAEA-385 certification measurements (red) and FRNA analysis (green) for individual emission lines of $^{226}$Ra and the $^{222}$Rn daughters $^{214}$Pb and $^{214}$Bi.**

The IAEA-385 material is the most rigorously-calibrated soil standard available to date and critical to our ability to validate high-quality gamma spectrometry (International Atomic Energy Agency, 2013; Pham et al., 2008). Even so, the original recommended $^{226}$Ra activity of 22.7 ±0.7 Bq kg$^{-1}$ (Pham et al., 2005) is 3.6% higher than the eventual certification value of 21.9 Bq kg$^{-1}$. The lower value derives from the





averaging of direct $^{226}$Ra measurements with lower estimates from $^{214}$Pb (352 keV) and $^{214}$Bi (609 keV)
(Fig. 8b-e). The latter should be viewed as suspect since both Rn emanation and coincidence summing
losses can bias the $^{222}$Rn daughters too low. FRNA measurements appear to show a systemic $^{222}$Rn loss of
3.4% as the difference between $^{226}$Ra activities (22.7 ±0.3) and $^{222}$Rn (21.98 ±0.07) [p=0.0186].

For sample analysis we sought to confirm the accuracy of our $^{226}$Ra measures and therefore that Rn
disequilibrium is due to Rn loss and not an analytical artifact (Syam et al., 2020). We performed an
experiment with NIST-traceable $^{226}$Ra standard solution (Eckert and Ziegler; certified uncertainty <2%).
which is important to verify that $^{226}$Ra is measured accurately and with high enough activity to measure
it with percent-level precision (here approximately 360 Bq kg$^{-1}$). A second calibrated $^{228}$Ra solution was
prepared by dissolution of thorium ore OKA-2. Accuracy of both Ra source calibrations was previously
confirmed by isotope-dilution thermal ionization mass spectrometry (Landis et al., 2018). Briefly, both
the $^{226}$Ra and $^{228}$Ra standard solutions were co-precipitated in a wet synthesis of $MnO_2$ from $KMnO_4$ and
$MnCl_2$ at pH ~9. To replicate a typical soil sample in 110 cm$^3$ petri geometry, the synthesized Ra-$MnO_2$
was air dried onto a bed of quartz sand. The Ra-Mn-sand thus produced was packed into our standard
polypropylene petri as for a soil sample.


The petri was first closed with electrical tape and measured serially to observe Rn ingrowth under our
typical sample preparation. Yields of both $^{226}$Ra and $^{228}$Ra are within error of 100% of certified or
expected values which confirms robust preparation of the test material (Fig. 9). Short-lived $^{220}$Rn (aka
thoron, half-life =56 seconds) is fully retained by the sample, but grows in following the half-life of
intermediary $^{224}$Ra (3.6 days) which is expelled from the initial suspension of $MnO_2$ by alpha recoil.

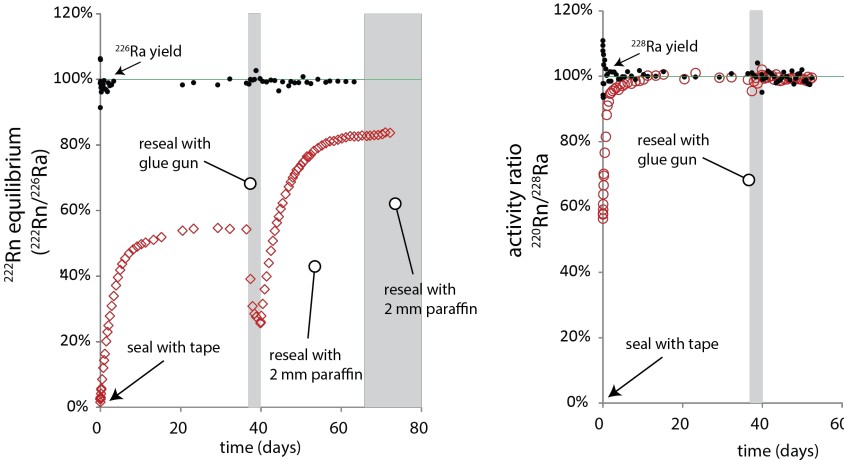

**Figure 9. Experimental confirmation of $^{222}$Rn emanation from 110 cm$^3$ soil laboratory samples, (a) $^{222}$Rn and $^{226}$Ra**
**source, (b) $^{220}$Rn and $^{228}$Ra source. Closed symbols show yield of $^{226}$Ra or $^{228}$Ra from the experiment, open**
**symbols show corresponding Rn yield or equilibrium as the Rn/Ra activity ratio.**

For $^{222}$Rn we observed an equilibrium ƒRn of just 55% after approximately thirty days. The sample was
then resealed using hot glue, but this reduced Rn retention, presumably, due to porosity in the glue that
visibly forms upon cooling. Finally, the sample was encased in 2 mm of paraffin wax. This final step
increased ƒRn to 85%, but the impracticality of the method coupled with imperfect Rn retention is a fatal
strike against its routine use in our laboratory.



Our subsequent efforts to seal samples under vacuum in aluminized Mylar bags improves Rn retention up to 90-95% for our 110 cm³ petri (Mauring & Gäfvert, 2013), but we have not standardized this approach due to remaining uncertainty in precise quantification of retention, the creation of additional laboratory waste and, in the face of these problems, the uncertain value of improving the indirect measure of $^{226}$Ra to defining $^{210}$Pb$_{ex}$ (see below).

**4. Defining excess Pb-210 ($^{210}$Pb$_{ex}$).**
FRN age models are based on the atmospheric component of $^{210}$Pb, which in soils and sediment must be separated mathematically from $^{210}$Pb that is supported by *in situ* production from U-series decay in soil minerals. By convention atmospheric $^{210}$Pb is measured as the excess over that supported ($^{210}$Pb$_{s}$) by its long-lived radioactive precursor $^{226}$Ra, i.e., $^{210}$Pb$_{ex}$ = $^{210}$Pb minus $^{226}$Ra. Due to analytical constraints
described above, $^{226}$Ra is routinely measured indirectly using $^{222}$Rn with the assumption that equilibrium is maintained between the two. We have already shown this is not accomplished in typical sample preparation.

Further, however, is the problem of Rn emanation from soils *in situ*: the very process that produces
excess $^{210}$Pb in the atmosphere yields a $^{210}$Pb deficit in soil. This means that supported $^{210}$Pb in soils is likely to be *overestimated* if measured by $^{226}$Ra. This leaves two open questions, (1) what is the correct metric for supported $^{210}$Pb *in situ*, and (2) how do we measure this in routine data collection in the laboratory? To clarify these problems, we introduce the term Pb enrichment, $\varepsilon$Pb, which is the ratio of $^{210}$Pb to $^{222}$Rn in deep soil samples believed free of atmospheric $^{210}$Pb. Importantly, for all soil pits we
have measured $\varepsilon$Pb =102 ±3% (±RSE, *n*=30) which means that supported $^{210}$Pb is typically in equilibrium with measured $^{222}$Rn. This is best illustrated in the soil depth profile where total $^{210}$Pb converges asymptotically with $^{222}$Rn, whereas both are overestimated by $^{226}$Ra (Fig. 10). Interestingly, this implies that Rn loss from the measured sample is comparable to loss in the soil *in situ*, and, as observed above, that Rn emanation from standard laboratory samples relates to a physical property of the soil itself.


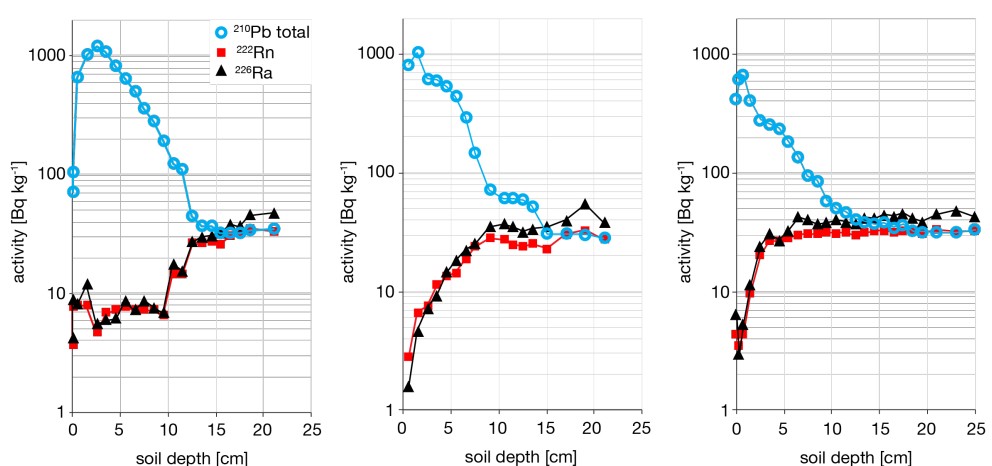

**Figure 10. Defining excess $^{210}$Pb in soils. (a) Hubbard Brook Experimental Forest (HuB01), (b) Experimental Lakes Area soil 302-1, (c) Harvard Forest LTER (HaF01). Total $^{210}$Pb typically converges with $^{222}$Rn at depth which indicates that $^{222}$Rn is an appropriate estimate of supported $^{210}$Pb.**


We therefore conclude the following regarding the definition of $^{210}$Pb$_{ex}$ in soils,



1. $^{226}$Ra is not an appropriate direct measure of supported $^{210}$Pb due to chronic emanation $^{222}$Rn from the soils *in situ*, which results in large $^{210}$Pb deficiencies in deep soils which are isolated from deposition of atmospheric $^{210}$Pb$_{ex}$.


2. *in situ* $^{222}$Rn emanation in the soil profile can be assessed directly by gamma measurements, as an asymptotic difference between $^{226}$Ra and $^{210}$Pb in deep soils.

3. measure of $^{222}$Rn by daughters $^{214}$Pb and $^{214}$Bi provides excellent precision and reproducibility among multiple emission lines, 5-10 times better than possible with direct measure of $^{226}$Ra.

4. Rn emanation from the laboratory sample is both pronounced and yet systematic despite sealing of containers and incubating >18 days following standard practice.


5. nonetheless, $^{222}$Rn measured in the sample typically converges with $^{210}$Pb, or is asymptotic with $^{210}$Pb, suggesting that $^{222}$Rn measure in the laboratory sample is the best available option for defining $^{210}$Pb$_{ex}$ in soil.

For defining $^{210}$Pb$_{ex}$ we use the following expression, noting that this accommodates cases where $\varepsilon$Pb is not equal to 1, by assuming a constant emanation rate throughout the soil profile (Landis et al., 2016):

$$^{210}\text{Pb}_{ex} = {}^{210}\text{Pb}_T - {}^{222}\text{Rn}(\varepsilon\text{Pb}) \tag{1}$$

Propagated uncertainties in $^{210}$Pb$_{ex}$ when calculated by Eq. 1 are shown in Fig. 11.

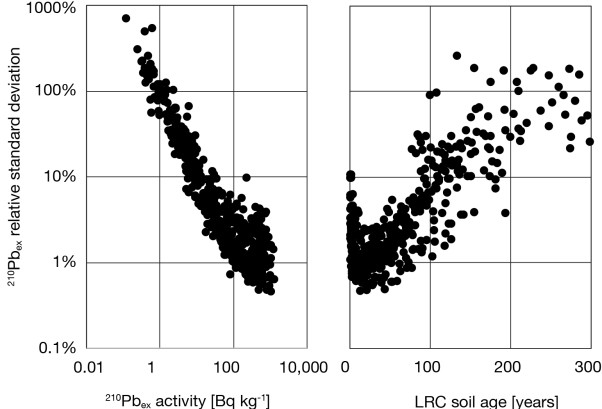

**Figure 11. Propagated uncertainties for $^{210}$Pb$_{ex}$ as function of (a) total $^{210}$Pb$_{ex}$ activity, (b) LRC soil age.**

### 5. Demonstrating LRC concordance with bomb pulse $^{241}$Am

Demonstrating concordance of LRC ages with independent chronometers is important for verifying both the suitability of soils for chronometry and veracity of data production (Landis, Obrist, et al., 2024). We report $^{241}$Am activities for this purpose, but it should be noted that peak bomb-pulse deposition is indicated at the layer of maximum $^{241}$Am density (Bq m$^{-2}$ cm$^{-1}$) rather than activity concentration since the latter is skewed by changes in bulk density especially at the soil surface. $^{241}$Am density is readily

calculated from the data provided. In some cases, the shape of $^{241}$Am bomb-pulse peak can be grossly distorted due to acceleration and deceleration of colloid migration in the soil profile (Landis, Taylor, et al., 2024), and in these cases the cumulative $^{241}$Am curve becomes an alternate tool for identifying peak bomb fallout since the history of $^{241}$Am in the atmosphere is well known (Appleby et al., 1991; Health and Safety Laboratory, 1977).




**6. Assessing uncertainties on radionuclide measurements.**

Our reported uncertainties for radionuclide activities are internal estimates which are propagated from uncertainties on peak integration, which includes background subtraction and deconvolution.

Uncertainties due to efficiency calibration and self-absorption correction are not included because, in part, we have not established the degree to which these may be autocorrelated among photo energies. However, external precision can be estimated conservatively by propagating an additional 2% uncertainty as described in *Sect. 2.3*.

*6.1. Z-scores to assess uncertainty estimates*

The accuracy of reported integration uncertainties can be assessed using multiple emission lines for a single radionuclide, since each must conform to the same true mean activity value. We show this by means of *Z*-score, which is the difference between emission line activities, normalized by the propagated uncertainty for the difference. The standard deviation of *Z* should be equal to 1 if the estimated

integration uncertainties accurately represent the true distribution of random error in the measurements (Fig. 12). We observe standard deviations of 0.8, 1.2, and 1.4 for $^{222}$Rn, $^{228}$Th, and $^{228}$Ra, respectively. These are close to 1 which implies that the integration uncertainties are robust. In the case of $^{228}$Th and $^{228}$Ra the real uncertainties are somewhat underestimated which means that uncertainties in calibration contribute an additional error, and one that is similar in scale to uncertainties in

integration, i.e., if similar errors are propagated the result is a 40% higher than either, $\sqrt{(1^2+1^2)}=1.4$.

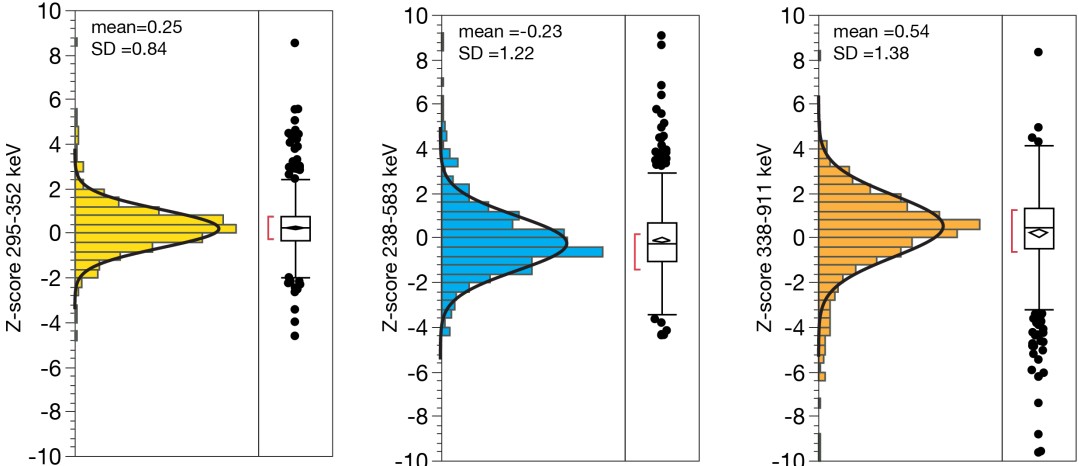

**Figure 12. Distribution of *Z*-scores for multiple emission lines of (a) $^{222}$Rn (295, 352 keV); (b) $^{228}$Th (238, 583 kev); (c) $^{228}$Ra (338, 911 keV). *Z* expresses the difference between two measures in units of σ and is calculated as the**

**difference between two photopeak measures of activity, divided by propagated uncertainty of the two measures.**

On the other hand, our prior assessment suggests that integration uncertainties are overestimated by on the order of 30%, based on the unique experiment of measuring radioactive decay of $^7$Be since the true

value of the sample is known based on its rate of decay, and estimated uncertainties can be evaluated against the observed dispersion of measurements around this known value (Landis et al., 2012). Overestimation of integration errors is consistent with our observed *Z* <1 for the $^{222}$Rn lines comparison above. That uncertainties for $^{228}$Th and $^{228}$Ra are greater than for $^{222}$Rn suggests that the interaction of coincidence-summing and self-absorption effects yields higher error in calibration, since these are not

handled independently in our calibration routine, i.e., by using a Th point source instead of U for Th-





series photopeaks. Future work is required to determine whether one $^{228}$Ra measurement may be preferred over others (e.g., among the $^{228}$Ac photopeaks at 209, 338, and 911 keV).

### 7. Spectrum accuracy evaluated using $^{228}$Th:$^{228}$Ra

The $^{228}$Th:$^{228}$Ra ratio is an important metric for evaluating accuracy of gamma measurements since it can generally be assumed to have an equilibrium value of 1. To ensure this is true we omit samples with LRC ages <20 years or $^{228}$Ra <10 Bq kg$^{-1}$ since these may have true $^{228}$Th:$^{228}$Ra disequilibrium due to variable uptake of Th and Ra during plant growth (Landis, 2023). The mean $^{228}$Th:$^{228}$Ra ratio for the data set is 1.015 ±0.038 SD, ±0.002 SE ($n$=390), showing a minor bias of 1.5%. Importantly, the standard deviation

for Z (for the difference between $^{228}$Th:$^{228}$Ra and 1) = 1.20, which shows that the estimates of uncertainties are reasonable (Fig. 13).

**Figure 13.** Assessing analytical accuracy using the measured $^{228}$Th:$^{228}$Ra ratio in soils. (a) measured $^{228}$Th:$^{228}$Ra ratio for all soils, (b) *Z*-score for difference between measured ratio and 1.


### 8. Certified Reference Materials

Availability of a few high-quality certified reference materials provides a means for external validation of environmental gamma data. We recommend IAEA-385 (sediment), IAEA-447 (moss soil), and NRC CLV-1 (spruce twig), since these have rigorous quality control and uncertainties low enough (typically <5%) to

allow verification of high-precision methods. Performance of our methods against these standards for radionuclides in addition to $^{226}$Ra is reported elsewhere with errors against certified values <5% (Landis et al., 2012, 2016).

### 9. Detection limits

Detection limit applies to total radioactivity in the detector shield [Bq] which is the product of mass and concentration, which means that it is dependent on sample bulk density since the sample volume is fixed. Detection is also inversely dependent on sample count time following Poisson counting statistics. Detection is also influenced by composition of the sample with respect to other elements and radionuclides because these influence photopeaks backgrounds through Compton scattering. Because

detection limits vary based on sample mass, composition, and count time they are not easily compiled and compared. We instead defer to reported uncertainties so that the value or weight of each datum can be independently assessed by users on this basis. Given this caveat, estimated detection limits for four-day count times, 100-gram samples, and approximately crustal-average U and Th content are approximately as follows: $^{7}$Be (0.5 Bq kg$^{-1}$); $^{210}$Pb (1.5 Bq kg$^{-1}$); $^{137}$Cs (0.3 Bq kg$^{-1}$); $^{241}$Am (0.15 Bq kg$^{-1}$);

$^{186}$Ra (2.5 Bq kg$^{-1}$); $^{222}$Rn (0.2 Bq kg$^{-1}$); $^{228}$Th (0.2 Bq kg$^{-1}$); $^{228}$Ra (1.0 Bq kg$^{-1}$);





## 10. Data versions, omissions, and error codes in the database

Multiple data reduction generations are represented in the database, with notable differences being the introduction of new photopeaks such as 209, 338, and 583 keV in later versions. Data fields are empty in preceding analyses.

In some rare cases, samples that are unable fill a 110 cm³ petri are run instead in 10 cm³ petris, with the lower volume incurring a corresponding loss in data quality due to low total activity and resulting counting statistics. Because data quality for the smaller 10 cm³ petri is poor and export fields are not aligned with data reduction for the standard 110 cm³ petri, these data are not reported in the database.

The error #NUM! is a peak integration error typically resulting from error in uncertainty estimation for integration of small peaks that are near detection. This can propagate through deconvolution regimes, i.e. error in uncertainty of $^{228}$Ac integration will produce error in uncertainty for $^7$Be.

## 11. Reported data fields

The reported data are tabular in form. Row one is data identifier 1. Row two is data 2, and row three is units or identifier 3. Subsequent rows list sample data per sample. Reported data include the following:

1. pit code: unique identifier for the soil pit.
2. sample name: unique identifier for the soil pit depth interval.
3. sample description: including horizon and/or depth interval.
4. horizon code: soil taxonomic horizon description.
5. interval mass [kg m$^{-2}$]: total mass of fine soil fraction (<2 mm) for the collected depth interval.
6. soil depth [cm]: starts at soil surface =0 rather than at mineral soil surface.
7. $R_{acc}$; accumulative $^7$Be:$^{210}$Pb ratio for overlying soil layers used in implementation of LRC model.
8. propagated uncertainty (1-*sigma*) for $R_{acc}$.
9. relative standard deviation (**RSD**) of $R_{acc}$.
10. count time [seconds] for collected gamma spectrum.
11. activity concentrations [Bq kg$^{-1}$] for the following radionuclides, with their propagated internal uncertainties: $^{210}$Pb (46 keV), $^7$Be (477 keV), $^{241}$Am (60 keV), $^{137}$Cs (662 keV), $^{234}$Th (63 keV), $^{226}$Ra (186 kev), $^{222}$Rn (295, 352, 609 keV), $^{228}$Th (238, 538 keV), $^{228}$Ra (338, 911 keV), $^{40}$K (1460 keV).

## 12. Data availability statement

The data described in this manuscript are freely available to the public (https://doi.org.10.17632/cfxkpn6hj9.1; Landis, 2025).

## 13. Conclusions

The central limitation to advancing chronometry of modern terrestrial biogeochemistry, including atmosphere-soil interaction and soil processes, is availability of required radionuclide data. We hope that by providing a large, open-access dataset we can help catalyze interest among new research groups and funding agencies in making new investments towards similar data acquisition efforts.

## 14. Competing Interests

The authors declare that no competing interests influence this work.





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
