# Peer review of "Data for Modern Soil Chronometry using Fallout Radionuclides"

_Earth System Science Data, 2025_

## Author Response (AR1)

**Reviewer #1**

Overview: This manuscript describes a large (>700 samples), geographically diverse (>35 locations) publicly available dataset of radionuclides measured in soils by gamma detection with the goal of improving studies that use modern chronometry. This work will be of great value to those seeking to set up and/or improve the analytical rigor of their gamma spectrometry facilities. The work is clear and thorough, and I have only minor edits listed below.

Minor [I provide some copy edits because these can sometimes, frustratingly, be missed at the copy edit stage]

> Thanks to the Reviewer for a careful and considered suggestions.

Line ~93: accomplishes

> Changed as requested.

Line ~111: photopeak energy in the ??? (described below)

> Now reads "each photopeak energy".

Line 165: important

> Changed as requested.

Line ~178: notable

> Changed as requested.

Line 185: measurable

> Change to "measurement".

Line ~247: I believe Pb-214 comes before Bi-214. Do the >> indicate missing steps? Is this a common notation?

> We have clarified that >> means omitted radionuclide, which we have not seen elsewhere but we think useful in the context of what can be measured in the gamma spectrum.

Section 4: One reason I was taught for measuring Rn-222 daughters Pb-214 and Bi-214 throughout the Pb-210 depths rather than relying on a measure of Pb-210 beyond a

depth at which excess and supported converge, is to account for potential stratigraphic changes in lithology, grainsize, organic matter content, etc. Perhaps this is less of a concern in soils than coastal sediments but perhaps it is worth noting in section 4.

> This is an excellent point to emphasize in Section 4, it is very important in soils since where strong mineralogical changes typically occur due to chemical weathering. We write "For defining 210Pbex at each soil depth we use the following expression. The calculation uses a direct measure of 222Rn at each depth to accommodate changes in soil organic matter and mineral abundance, but assumes a correction for Rn emanation rate for the full the soil profile (Landis et al., 2016)."

Line 460: end sentence on a period.

> Changed as requested.

Line ~468: unable *to* fill

> Changed as requested.

**Reviewer #2**

**General Assessment**

The manuscript presents a comprehensive and meticulously curated dataset of environmental gamma spectrometry measurements for soil chronometry using fallout radionuclides (FRNs), specifically 77Be and 210210Pb, alongside ancillary isotopes. The work is highly relevant to biogeochemical, environmental, and geochronological research, offering a valuable resource for quantifying soil processes over decadal to centennial timescales. The methodological rigor, clarity of data presentation, and open-access approach are commendable. However, several aspects require clarification or improvement to enhance reproducibility, accessibility, and interpretation.

> Thanks to the Reviewer for careful and construction suggestions!

**Data Accessibility and Documentation**

While the dataset is hosted on Mendeley Data, the manuscript would benefit from a more detailed description of the data structure (e.g., file formats, column headers, metadata).

The dataset structure is now detailed in Section 11, with descriptions of both rows (including headers), and columns (both metadata and data).

Provide explicit instructions for users to navigate potential pitfalls (e.g., handling of error codes like #NUM!, treatment of small-volume petri dish samples).

Line 485: We have recommended replacing #NUM! error code with 100% relative uncertainty, "For purposes of error propagation the error can reasonably be estimated by assuming 100% relative uncertainty, i.e., equal to the measurement value."

Line 480: We specified that data from the smaller 10 $cm^3$ geometry are not included in the database at present, so no need for users to deal with this issue. We have edited to read, "In some rare cases (e.g., deep peat cores), samples that are unable fill a 110 $cm^3$ petri are run instead in 10 $cm^3$ petris. The lower volume incurs a commensurate loss in data quality due to low total activity and resulting counting statistics. Data quality for the smaller 10 cm3 petri is poor and their use is rare. These data are not reported in this database at the present time."

**Uncertainty Quantification**

The exclusion of efficiency calibration and self-absorption uncertainties from reported radionuclide activities (Sect. 6) may underestimate total error. Justify this choice or include a consolidated uncertainty budget.

Line 405: We believe the internal (integration) estimate provides the best direct assessment of measurement quality because it is related directly to radionuclide activity and underlying Poisson statistics. The additional uncertainties from calibration are both fixed and low, and easily propagated as a second, final step assuming 2%. This is now clarified as follows, "These provide a direct assessment of measurement quality related to total radionuclide activity and related Poisson statistics. Uncertainties due to efficiency calibration and self-absorption correction are not included in reported uncertainties within the dataset because they are fixed and therefore easily consolidated in a second step by propagating an additional 2% uncertainty as described in Sect. 2.3. We have not established the extent to which calibration uncertainties may autocorrelated across different radionuclide energies. These external uncertainties are assessed here for user consideration and will be appropriate, for example, when comparing measurements in this dataset to those from other laboratories."

Clarify whether the 2% external precision estimate (Sect. 2.3) is propagated in the final dataset or intended for user consideration.

As above, it is included for user consideration.

**Rn Emanation and 210210Pbex Definition**

The discussion of Rn loss (Sect. 3.5–4) is critical but could be streamlined. Consider consolidating Figs. 7–10 to highlight the key conclusion:  222Rn (not 226Ra) is the preferred proxy for supported 210Pb due to systematic Rn emanation.

Beginning Line 245: We appreciate the recommendation. The issue of Rn loss may be controversial so believe important to retain all Figs. 7-10. However, we have tried to better focus the discussion to keep the primary point at the fore, as the Reviewer has recommended,

Explicitly state whether the proposed 210Pbex calculation (Eq. 1) is implemented in the provided dataset or requires user derivation.

Line 375: We have clarified that the 210Pbex calculation is not implemented within the dataset. "Calculation of 210Pbex is not provided in the dataset but can be performed following Eq. 1."

**Methodological Reproducibility**

The semi-automated peak integration routine (Sect. 2.4) relies on Excel. Provide a script or flowchart to ensure transparency, given Excel's limitations for complex spectral analysis.

Line 132...The calculation is actually quite simple and is well described in the referenced source (Landis et al. 2012). We do not run script for this but have now provided the calculations here as well as a flow chart for data reduction as new Fig. 2.

Detail the criteria for omitting samples with 228Ra <10 Bq kg−1 or ages <20 years (Sect. 7) to avoid potential bias in 228Th: 228Ra equilibrium assessments.

We have clarified that we omit samples from this assessment with either $^{228}$Ra <10 Bq kg-1 (leaves and Oi horizon) or LRC ages <20 years (approximate horizon for $^{228}$Th:$^{228}$Ra steady-state).

**Bomb-Pulse Validation**

The divergence between 241Am and 137Cs profiles (Fig. 6) underscores the need for clearer guidelines on when 137Cs may still be useful despite mobility. Add a decision tree or table summarizing preferred chronometers under different soil conditions.

Line 225: Unfortunately, we do not have reliable guidelines at this time due to lack of sufficient data. We have added a caution to suspect this "especially where soil K may be deficient" after Kaste et al. 2021. We also suggest Cs may be used only "if it can be shown, e.g. by regression, that there is no systematic bias between the 241Am and 137Cs depth profiles or mass distributions."